

# Using *Peromyscus leucopus* as a biomonitor to determine the impact of heavy metal exposure on the kidney and bone mineral density: results from the Tar Creek Superfund Site

Maha Abdulftah Elturki[1,2,3]

[1] Department of Environmental Sciences, Oklahoma State University, Stillwater, Oklahoma, United States
[2] Department of Integrative Biology, Oklahoma State University, Stillwater, Oklahoma, United States
[3] Zoology Department, Faculty of Science, University of Benghazi, Benghazi, Libya

Corresponding author
Maha Abdulftah Elturki,
elturki@okstate.edu

## ABSTRACT

**Background:** Human population growth and industrialization contribute to increased pollution of wildlife habitats. Heavy metal exposure from industrial and environmental sources is still a threat to public health, increasing disease susceptibility. In this study, I investigated the effects of heavy metals (cadmium (Cd), lead (Pb), and zinc (Zn)) on kidney and bone density.

**Objective:** This study aims to determine the concentrations of Cd, Pb, and Zn in soil and compare them to the levels of the same metals in *Peromyscus leucopus* kidney tissue. Furthermore, the study seeks to investigate the impact of heavy metals on bone density and fragility using the fourth lumbar vertebra (L4) of *P. leucopus*.

**Methods:** Cd, Pb, and Zn concentrations in soil specimens collected from Tar Creek Superfund Site (TCSFS), Beaver Creek (BC), and two reference sites (Oologah Wildlife Management Area (OWMA) and Sequoyah National Wildlife Refuge (SNWR)). Heavy metal concentrations were analyzed using inductively coupled plasma-mass spectroscopy (ICP-MS). Micro-computed tomography (µCT) was used to assess the influence of heavy metals on bone fragility and density.

**Results:** On the one hand, soil samples revealed that Pb is the most common pollutant in the sediment at all of the investigated sites (the highest contaminated site with Pb was TCSFS). Pb levels in the soil of TCSFS, BC, OWMA, and SNWR were found to be 1,132 ± 278, 6.4 ± 1.1, and 2.3 ± 0.3 mg/kg in the soil of TCSFS, BC and OWMA and SNWR, respectively. This is consistent with the fact that Pb is one of the less mobile heavy metals, causing its compounds to persist in soils and sediments and being barely influenced by microbial decomposition. On the other hand, the kidney samples revealed greater Cd levels, even higher than those found in the soil samples from the OWMA and SNWR sites. Cd concentrations in the kidney specimens were found to be 4.62 ± 0.71, 0.53 ± 0.08, and 0.53 ± 0.06 µg/kg, respectively. In addition, micro-CT analysis of L4 from TCSFS showed significant Pearson's correlation coefficients between Cd concentrations and trabecular bone number (−0.67, $P \leq 0.05$) and trabecular separation (0.72, $P \leq 0.05$). The results showed no correlation between bone parameters and metal concentrations at reference sites. This study is one of
the few that aims to employ bone architecture as an endpoint in the field of biomonitoring. Furthermore, this study confirmed some earlier research by demonstrating substantial levels of heavy metal contamination in soil samples, kidney samples, and *P. leucopus* L4 trabecular bone separations from TCSFS. Moreover, this is the first study to record information regarding bone microarchitecture parameters in *P. leucopus* in North America.

## INTRODUCTION

The Tar Creek Superfund site (TCSFS), Ottawa, OK is in northeastern Oklahoma near the Kansas-Oklahoma border. TCSFS covers a 40-square-mile area and is one of the Tri-State Mining District (Oklahoma, Kansas, and Missouri) sites (*Bigby & Jim, 2021*; *Coffin et al., 2022*; *Hayhow, 2021*; *McCann & Nairn, 2022*; *Park et al., 2020*). These sites include territories of ten tribal nations such as the Quapaw Nation and several other communities such as Picher, Cardin, North Miami, and Commerce (*Agency for Toxic Substances & Disease Registry, 2015*). Lead and zinc ores were mined at TCSFS from the early 1900s to the late 1970s.

*Peromyscus leucopus* has been the subject of several different studies on the effects of environmental contaminants (*Beyer et al., 2018*; *Johnson et al., 2021*; *Rubino, Oggenfuss & Ostfeld, 2021*).

The Agency for Toxic Substances and Disease Registry (ATSDR) lists the major pathways of exposure to lead contamination at TCSFS as contaminated air, contaminated water, contaminated food resources, and contaminated soil. Such human health problems as respiratory illness, liver dysfunction, and reproductive and renal failure can occur after exposure *via* these pathways (*Agency for Toxic Substances & Disease Registry, 2015*; *Briffa, Sinagra & Blundell, 2020*).

Heavy metals such as cadmium, lead, and zinc at TCSFS were studied because of their potential effects on human health and their accumulation in small mammals' bodies and other terrestrial fauna (*Briffa, Sinagra & Blundell, 2020*; *Martín et al., 2021*; *Monchanin et al., 2021*; *Sánchez-Chardi & Nadal, 2007*; *Sánchez-Chardi et al., 2007*). Numerous human health issues have been documented after exposure to cadmium because of its ability to substitute for other metals and nutrients such as zinc (*Beyersmann & Hartwig, 2008*; *Briffa, Sinagra & Blundell, 2020*).

Depending on the concentration, duration, and species of the metals involved, exposure to metals has a detrimental effect on bone health and may lead to osteoporosis and an increased risk of fracture. Lead and certain other metals are stored permanently in bones, which in mammals and birds may hold 90% of the total body burden (*Bjørklund et al., 2020*). Furthermore, lead, cadmium, and chromium may have a non-redundant function in the pathogenesis of osteoporosis at the cellular/molecular and epigenetic levels (*Scimeca et al., 2017*).

The most common contaminants in the soil are cadmium as a heavy metal and atrazine as an organic pollutant, both can be considered potential pathogenic pollutants (*Said et al., 2022*; *Scimeca et al., 2017*). Small amounts of free cadmium ions are more toxic than bound cadmium ions, and cadmium can cause toxicity in different organs including the pancreas, testis, and nervous system. Elimination of cadmium by the kidneys is slow. Chronic exposures to cadmium ions can result in proteinuria and tubular dysfunction in the proximal tubules (*Davenport, 2020*; *Godt et al., 2006*; *Yan & Allen, 2021*; *Zavala-Guevara et al., 2021*). Renal toxicity from cadmium exposure is correlated with the number of cadmium ions in kidney tubule cells, reabsorption, degradation of cadmium metallothionein complexes, and excess production of metallothionein by renal tubules (*Davenport, 2020*; *Zavala-Guevara et al., 2021*). Cadmium toxicity observed in the renal cortex of laboratory rats (*Rattus rattus*) resulted in cytosolic damage and renal malfunction after oral administration of cadmium chloride ($CdCl_2$) (*Alimba et al., 2021*; *Hashim et al., 2018*; *Siddiqui, 2010*).

Lead is a non-essential metal and is one of the abundant toxic metals at TCSFS. Lead exposure can result in acute and chronic toxic effects. Lead accumulates in wild mammals in tissues such as the kidney, liver, and bone. In smelter and mining sites, lead and zinc were recorded in wood mice (*A. sylvaticus*), bank voles (*C. glareolus*), and field voles (*M. agrestis*), with high lead concentrations in bones; 42–68% of total lead found in body tissues was contained in bone (*Bjørklund et al., 2020*; *Ecke et al., 2020*; *Jasiulionis et al., 2018*; *Turna Demir & Yavuz, 2020*).

Lead exposure decreases bone mineral density (BMD) which can cause osteoporosis (*Bjørklund et al., 2020*; *Campbell et al., 2004*; *Scimeca et al., 2017*). In addition, lead exposure inhibits osteoblast function. Zinc is another trace metal that was measured in this study. Zinc has important functions in bone formation, turnover, and metabolism (*Bekheirnia et al., 2004*; *Ceylan, Akdas & Yazihan, 2021*; *Gaffney-Stomberg, 2019*). Zinc as a cofactor plays essential roles in other tissue and enzyme functions that are important for bone mineralization and development such as alkaline phosphate and collagenase (*Bekheirnia et al., 2004*; *Ceylan, Akdas & Yazihan, 2021*; *Gaffney-Stomberg, 2019*).

Bone as a connective tissue has different sizes, shapes, and structures that serve important functions. Mineralized bone is the osseous tissue that gives bone rigidity. Bone tissue accumulates heavy metals such as cadmium and lead. Furthermore, bone tissue is one of the tissue markers that indicate xenobiotic and metal exposure (*Bjørklund et al., 2020*; *Scimeca et al., 2017*). *P. leucopus* as bioindicator species recorded as the most common species of the small mammals at TCSFS is the white-footed mouse (*Peromyscus leucopus*) (*Park et al., 2020*; *Phelps & McBee, 2009*; *Phelps & McBee, 2010*).

Studying bone microarchitecture helps to evaluate the toxic effects on bone after exposure to heavy metals. Bone dysfunction and osteoporosis are reported as toxic effects of exposure to cadmium (*Buha et al., 2019*; *Luo et al., 2021*; *Youness, Mohammed & Morsy, 2012*). A significant decrease in bone density and the presence of osteopenia has been recorded in women who are exposed to cadmium from environmental sources (*Engström et al., 2012*; *Jalili et al., 2020*; *Kim et al., 2021*).

Because soil is a valuable natural resource that may contain environmental contaminants, the concentrations of heavy metals (cadmium (Cd), lead (Pb), and zinc (Zn)) in TCSFS, BC, and reference sites were determined using soil samples. The presence of contaminants in a biological source is assessed using soil, and changes in the physiological parameters of *P. leucopus* as a biomonitoring species are investigated. As a result, the goal of this study is to compare the levels of Cd, Pb, and Zn in the soil to those found in *Peromyscus leucopus* kidney tissue. The study uses the fourth lumbar vertebra (L4) of *P. leucopus* to investigate the effect of heavy metals on bone density and fragility. As a result, we may have a better understanding of how these contaminants affect the natural resources of the environment as well as the associated fauna. Furthermore, one of the purposes of the study is to link biological factors to the presence of heavy metals in soil in a field setting. This study is one of the few that aims to employ bone architecture as an endpoint in the field of biomonitoring. Moreover, this is the first study to record information regarding bone microarchitecture parameters in *P. leucopus* in North America.

## MATERIALS AND METHODS

### Study sites

The study included three different sites. Tar Creek Super Fund Site (TCSFS) at Beaver Creek and two reference sites, Sequoyah National Wildlife Refuge (SNWR) and Oologah Wildlife Management Area (OWMA) (Fig. 1). Soil samples were collected from these sites for metal analysis (lead, cadmium, and zinc). The TCSFS has a large population of white-footed mice (*Peromyscus leucopus*) which were used in this study to analyze the influence of heavy metals on the physiological alteration in kidney tissues. Both SNWR and OWMA are good sites to compare with TCSFS because no mining has been recorded on these sites. The use of two reference sites rather than just one provides a more accurate comparison to the highly contaminated TCSFS. In addition, these two sites are geographically distant (Fig. 1). Other reasons to choose these sites were to compare relationships among soil metal levels, tissue metal levels, and metallothionein induction in *P. leucopus* species, which is present in all three sites. Position for collection sites was detected with the help of Eterx Vista CX Garmin and Google. Samples were collected to a depth of 18–20 cm from each position (Fig. 1). Field experiments were approved by the Oklahoma State University Institutional Animal Care and Use Committee (ACUP No. AS 066).

### Soil sampling

Soil samples were collected from the study sites following the procedure that is described by USEPA (*United States Environmental Protection Agency, 2005*). A random design was used to collect soil samples in each site separately. Moreover, a meter scale was used to measure the distances between soil samples. Eight duplicate soil samples were collected from each site. Each sample was labeled as T1-1, T1-2, T2-1, T2-2, *etc*. where T identifies TCSFS, the first numeral (1–8) indicates the number of the sample, and the second numeral (1–2) points to the original or the duplicate. Samples were collected from each
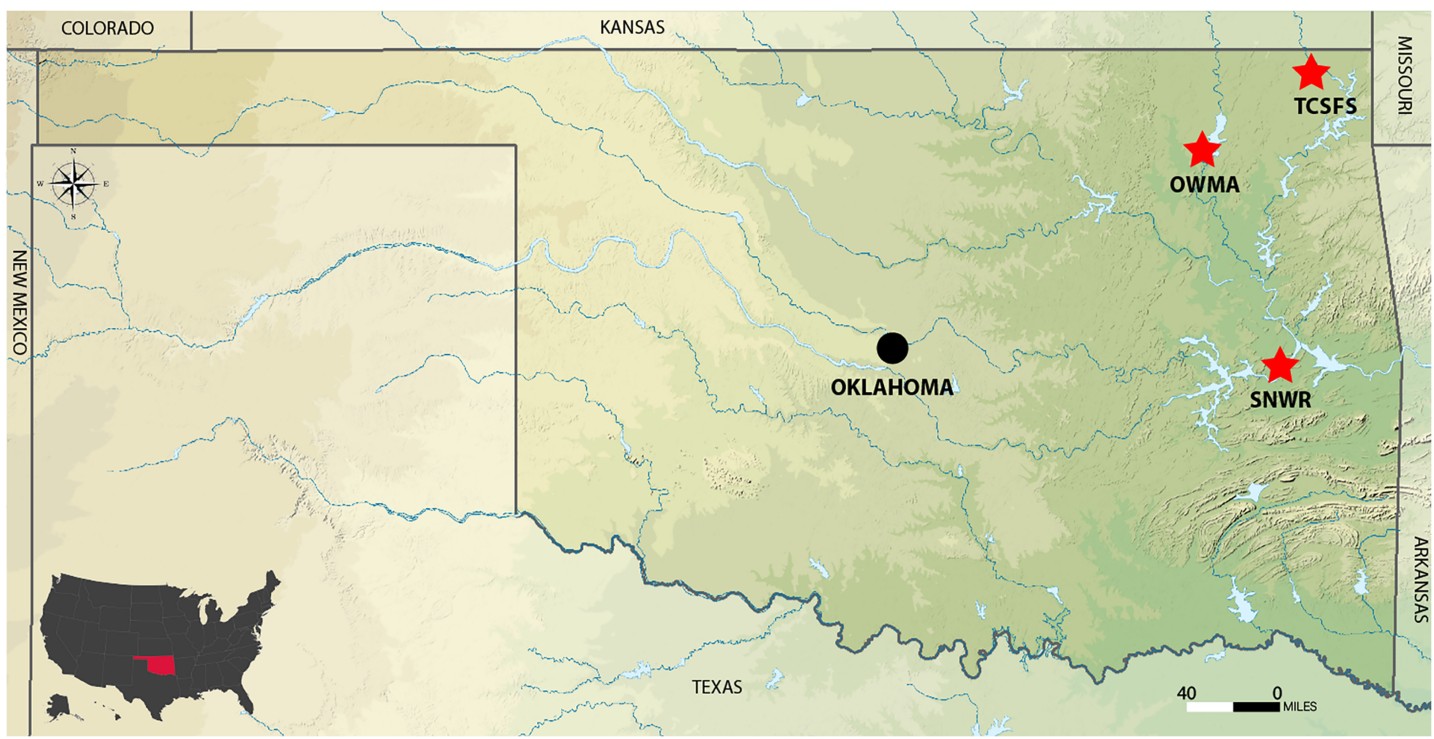

**Figure 1** Location of the Tar Creek Superfund Site and two reference study areas (red stars) in northeastern Oklahoma State, USA. **OWLA**, Oolaga wild life management area. **SNWR**, Sequoyah national wildlife refuge. **TCSFS**, Tar Creek Superfund Site. Adapted from Wikimedia Commons (https://upload.wikimedia.org/wikipedia/commons/1/16/USA_Oklahoma_relief_location_map.svg) by SANtosito, 3 February 2021. Creative Commons Attribution-Share Alike 4.0 International license.

position with a 10% HCl acid–washed metal scoop to a depth of 18–20 cm. Samples were homogenized and weighed separately. Soil samples were dried in separate 10% HCl acid-washed and labeled polypropylene plates at room temperature for 2 weeks. Also, dried soil samples were sieved twice using the first 1 mm sieve size, No. 18 (USA. Standard Test Sieve) followed by a second 250 μm opening sieve (Fisher Scientific Company, Pittsburgh, PA, USA). Samples were saved in labeled acid-washed glass bottles.

## Soil digestion

Soil samples were prepared in the trace mineral laboratory of the Nutrition Sciences Department using microwave digestion. All soil samples from Tar Creek Super Fund Site (TCSFS) and the two reference sites, Sequoyah National Wildlife Refuge (SNWR) and Oologah Wildlife Management Area (OWMA) were weighed separately using the same digital balance and labeled polypropylene plates. Soil samples were digested using microwave digestion protocol (Milestone Inc., Shelton, CT, USA) specified by USEPA Method Number 3051A (*Kingston et al., 1997*; *United State Environmental Protection Agency, 1986*) according to the EPA procedure. Following the protocol for microwave digestion, all soil sample sizes were 0.5 g. For all soil samples, 0.5 g of soil was put in the labeled microwave vessel. Double distilled trace element grade nitric acid (99.999%) was purchased from Fisher Scientific Company (Pittsburgh, PA, USA). Microwave vessels were transferred to the hood and 10 ml of $HNO_3$ was added to the soil sample in each vessel.

The soil and acid solution were swirled slowly to mix soil with acid. A Teflon cover was placed on the Teflon vessel and pushed down, and an adaptor was placed on the flat part of the Teflon cover. The Teflon indicator ring was placed on the cover and pushed shut. Vessels were then introduced into the polypropylene microwave rotor. Each indicator was closed tightly with a torque wrench. A thermocouple was placed into reference vessel No.1 that contained a blank laboratory (control) sample, the microwave door was closed, and the machine was switched on.

The program was set on the USEPA Method Number 5031A. Samples were heated for 50 min at 100 °C. Next, the segments were pulled gently from the microwave after 5 min because they were hot and opened slowly using a torque wrench to release the pressure from the vessels. Samples were transferred to the hood to open the vessel's cover and yellow acid fumes evaporated from the vessels. The acid solution from vessels 10 ml was poured into a plastic 15 ml tube purchased from VWR. Vessels were washed with 5 ml double distilled water, and water was added to the first tube. All samples were digested following the same process. The digested soil samples were centrifuged at 1,200 g for 10 min. The samples were decanted into a new tube (2nd tube) gently and slowly to avoid solution contamination. The first tube that contained the soil was disposed of separately. The second tube for each digested sample was labeled as stock that was used for metal analysis. A total of 0.2 ml of the digested soil solution in 10 ml total volume samples were diluted by adding double distilled water (DDW) to 0.20 ml sample solution in the new tube (3rd tube). This tube of each sample was used for the Inductively Coupled Plasma-Mass Spectroscopy (ICP-MS) instrument (Perkin Elmer) to analyze metal (Pb, Zn, and Cd) concentrations in each sample separately.

## ICP-MS analysis

Cd, Pb, and Zn concentrations in soil specimens from both contaminated and uncontaminated sites were determined by ICP-MS according to USEPA Method 6010 (*United States Environmental Protection Agency, 1996*). Terbium was used as an internal standard (Perkin Elmer, Shelton, CT, USA). The ICP-MS instrument was calibrated to ensure the stability and consistency of all results. Before running samples by ICP-MS, five standard dilutions for ICP-MS were diluted. Internal standard (Terbium) solution (Perkin Elmer, Shelton, CT, USA) and double distilled water (DDW) were used for calibration and dilutions. All sample volumes were run in ICP-MS as 0.1 μl sample diluted to 10 ml (DDW) and add 20 μl (IS). Sample tubes were vortexed for 20 s before analysis.

## Kidney sampling

Frozen kidney biopsies of *Peromyscus leucopus* were provided by the OSU Collection of Vertebrates, Department of Integrative Biology. The *P. leucopus* had been collected from TCSFS, BC, and the reference sites, OWMA and SNWR. The kidneys were already frozen at −80 °C. Kidney samples were saved in a liquid nitrogen tank before subsampling. The kidneys were cut in a plastic petri dish placed on wet ice using a stainless-steel scalpel to one-tenth of the whole tissue and saved in plastic microtubes. Next, kidney samples

were refrigerated at −40 °C prior to digestion for metal analysis. Oklahoma State University Animal Care and Use provided full approval for this research (Protocol AS056).

## Tissue digestion

Kidney samples (Kid) were weighed using a digital balance. Small microwave tubes were used for kidney sample digestion (Milestone Inc., Shelton, CT, USA) and USEPA Method Number 3051A (*United State Environmental Protection Agency, 1986*). Each sample was weighed and put in small microwave tubes; 1 ml of concentrated (99.999) $HNO_3$ was put in the tube and then 0.15 ml of $H_2O_2$ was added to the same tube. The program was set on USEPA Method Number 5031A. The microwave operation required 50 min to finish digestion at the temperature of 100 °C. Next, the vessels were pulled gently from the microwave after 5 min. The segments were opened slowly using a torque wrench to release the pressure from the vessels. Samples were transferred to the hood to open the vessel's cover, and yellow acid evaporated from the vessels.

## Metal analysis

To ensure the stability and consistency of all results, Cd, Pb, and Zn concentrations in kidney specimens were determined by inductively coupled plasma mass spectrometry (ICP-MS) instrument (Perkin Elmer) according to USEPA method 6010 (*United States Environmental Protection Agency, 1996*). Terbium was used as an internal standard (Perkin Elmer, Shelton, CT, USA). Before samples were run, five standard dilutions for ICP-MS were prepared. All sample volumes were analyzed by ICP-MS as 80 μl sample with 10 μl Internal standard (Terbium) solution (Perkin Elmer, Shelton, CT, USA) and complete volume by DDW up to 5 ml. Sample tubes were vortexed for 20 s before analysis by ICP-MS.

## Bone microarchitecture
### μCT analysis

Eight skeletons of *P. leucopus* from each site were provided from the vertebrate collection from the Department of Integrative Biology at OSU. The lumbar 2, 3, 4, and 5 vertebrae section of each skeleton was excised and the L4 was scanned using a high-resolution computed tomography system or micro-CT scanner (μCT 40; Scano Medical AG, Zurich, Switzerland). The fourth lumbar vertebra was detected in each skeleton sample and saved as a 3-D image. The trabecular bone in the 3-D images of L4 was contoured in a 300–400 μCT slide image. L4 slices were contoured every 10 slices beginning 10 slices below the detection of the spongiosa and ending 10 slices from the growth plate. The threshold for evaluation was set as 350 (grayscale, zero-1000) for all slides. The trabecular bone was contoured to measure the connectivity density (ConnD), trabecular separation (TbSp), trabecular thickness (TbTh) (mm), trabecular number (TbN) (mm-1), and trabecular volume (TV) as a percent of bone volume fraction (BVTV) (bone volume/tissue volume) for individual lumbar vertebra. The 3D images of the results were evaluated, and the data set was exported to evaluate and analyze the results.

**Table 1 Concentrations of zinc, cadmium, and lead in soil samples gathered from the study areas.**

| Mineral/Site | n | SNWR | OWMA | TCSFS |
|---|---|---|---|---|
| Zn (mg/kg) | 16 | 20.0 ± 1.9[b] | 52.6 ± 5.0[b] | 14,083.9 ± 1825.8[a] |
| Cd (mg/kg) | 16 | 0.06 ± 0.01[b] | 0.15 ± 0.03[b] | 48.04 ± 3.98[a] |
| Pb (mg/kg) | 16 | 2.3 ± 0.3[b] | 6.4 ± 1.1[b] | 1,132 ± 278[a] |

Notes:
  OWMA, Oologah Wildlife Management Area; SNWR, Sequoyah National Wildlife Refuge; TCSFS, Tar Creek Superfund site.
  Means in rows not sharing the same superscript are significantly different from each other ($P < 0.0001$).

**Table 2 Body weights and kidney lead, cadmium, and zinc concentrations of *Peromyscus leucopus* individuals gathered from study areas.**

| Variables (Mean ± SE) | TCSFS | OWMA | SNWR |
|---|---|---|---|
| BW (g) | 25 ± 0.8[a] | 24 ± 1.3a | 23 ± 1.2[a] |
| Kidney Pb (μg/Kg) | 0.57 ± 0.10[a] | 0.04 ± 0.01[b] | 0.50 ± 0.01[b] |
| Kidney Cd (μg/Kg) | 4.62 ± 0.71[a] | 0.53 ± 0.08[b] | 0.53 ± 0.06[b] |
| Kidney Zn (μg/Kg) | 23.1 ± 3.3[a] | 18.5 ± 3.8[a] | 28.4 ± 4.6[a] |
| n | 17 | 16 | 16 |

Notes:
  BW, body weight; OWMA, Oologah Wildlife Management Area; SNWR, Sequoyah National Wildlife Refuge; TCSFS, Tar Creek Superfund site.
  Means in rows not sharing the same superscript are significantly different from each other ($P < 0.0001$).

## Statistical analyses

This study examined soil and kidney samples collected from a contaminated site, TCSFS, BC, and compared them with reference sites (SNWR and OWMA). In addition to soil and kidney samples, bone parameters from the contaminated site were compared with reference sites. Metal concentrations in the kidney were correlated with bone parameters. Pearson's correlation coefficients were determined for all samples taken together and by individual sites; Proc GLM, Proc Corr, SAS, V 9.4 were used and values of $P < 0.05$ were taken as significant.

## RESULTS

### Soil heavy metal levels

Comparison of the heavy metal levels in the soil samples from the contaminated site (TCSFS, BC), and the reference sites SNWR and OWMA (Tables 1–5) showed that cadmium concentrations in the soil samples of TCSFS, BC, and the reference sites were sharply different. Mean concentrations of Cd, Pb, and Zn mg/kg in soil samples are presented in Table 1. In TCSFS, BC as a contaminated site, soil cadmium concentrations (mean ± SE) at 48 ± 04 mg/kg were recorded significantly higher than those at the two reference sites (0.06 ± 0.01, and 0.15 ± 0.03 mg/kg).

As expected, lead concentrations in the TCSFS, BC soil samples (1,132 ± 278 mg/kg) were higher ($P < 0.0001$) than in the two reference sites (2.3 ± 0.33, 6.4 ± 1.1 mg/kg). Zinc concentrations in TCSFS, BC (14,083 ± 1,826) were also much higher ($P < 0.0001$) than in the two reference sites (20 ± 2 and 53 ± 5 respectively) (Table 1).

**Table 3 Bone microarchitecture parameters of *Peromyscus leucopus* (n = 24) collected from the study sites.**

| Variables (mean ± SE) | TCSFS | OWMA | SNWR |
|---|---|---|---|
| TV | 1.27 ± 0.14[a] | 0.81 ± 0.06[b] | 0.10 ± 0.12[b] |
| BVTV | 0.16 ± 0.02 | 0.18 ± 0.01 | 0.15 ± 0.02 |
| ConnD | 238.16 ± 50.34[a] | 157.41 ± 12.55[b] | 132.81 ± 16.75[b] |
| TbN | 4.11 ± 0.38 | 4.07 ± 0.11 | 3.63 ± 0.12 |
| TbSp | 0.26 ± 0.03 | 0.24 ± 0.01 | 0.28 ± 0.02 |

Notes:
BVTV, Bone Volume/Total Volume (bone volume fraction); ConnD, Connectivity Density; OWMA, Oologah Wildlife Management Area; SNWR, Sequoyah National Wildlife Refuge; TbN, Trabecular Number; TbSp, Trabecular separation; TCSFS, Tar Creek Superfund Site; TV, Total Volume.
Means in rows not sharing the same superscript are significantly different from each other ($P < 0.0001$).

**Table 4 Pearson's correlation coefficients for trabecular bone microarchitecture parameters of L4 and kidney metal concentrations from *Peromyscus leucopus* (n = 8) collected from the Tar Creek Superfund Site (TCSFS).**

| Variables | 1 | 2 | 3 | 4 | 5 | 6 | 7 | 8 |
|---|---|---|---|---|---|---|---|---|
| BVTV | – | – | – | – | – | – | – | – |
| ConnD | 0.81* | – | – | – | – | – | – | – |
| TbN | 0.87** | 0.95*** | – | – | – | – | – | – |
| TbTh | 0.19 | −0.31 | −0.28 | – | – | – | – | – |
| TbSp | −0.82* | −0.89** | −0.98*** | 0.31 | – | – | – | – |
| Kid-Cd | −0.49 | −0.52 | −0.67* | 0.21 | 0.72* | – | – | – |
| Kid-Pb | 0.58 | 0.21 | 0.41 | 0.46 | −0.51 | −0.43 | – | – |
| Kid-Zn | −0.24 | −0.21 | 0.01 | −0.47 | −0.14 | −0.13 | 0.22 | – |

Notes:
BVTV, Bone Volume/Total Volume (bone volume fraction); ConnD, Connectivity Density; Kid, Kidney; TbN, Trabecular Number; TbSp, Trabecular separation; TbTh, Trabecular thickness.
* $P \leq 0.05$.
** $P \leq 0.005$.
*** $P \leq 0.0005$.

**Table 5 Pearson's Correlation coefficients for trabecular bone microarchitecture parameters of L4 and kidney metal concentrations from *Peromyscus leucopus* (n = 16) collected from Oologah Wildlife Management Area and Sequoyah National Wildlife Refuge.**

| Variable | 1 | 2 | 3 | 4 | 5 | 6 | 7 | 8 |
|---|---|---|---|---|---|---|---|---|
| BVTV | 0.02 | – | – | – | – | – | – | – |
| ConnD | −0.14 | 0.37 | – | – | – | – | – | – |
| TbSp | 0.17 | −0.52* | −0.83*** | – | – | – | – | – |
| TbN | −0.29 | 0.56** | 0.86*** | −0.97*** | – | – | – | – |
| Kid-Cd | −0.22 | 0.26 | 0.08 | −0.30 | 0.33 | – | – | – |
| Kid-Pb | −0.20 | −0.17 | 0.09 | −0.20 | 0.19 | 0.37 | – | – |
| Kid-Zn | 0.11 | −0.20 | −0.02 | −0.17 | 0.11 | 0.39 | 0.85*** | – |

Notes:
BVTV, Bone Volume/Total Volume (bone volume fraction); ConnD, Connectivity Density; Kid, Kidney; TbN, Trabecular Number; TbSp, Trabecular separation; TbTh, Trabecular thickness.
* $P \leq 0.05$.
** $P \leq 0.005$.
*** $P \leq 0.0005$.

### Kidney heavy metal levels

As for the kidney samples (μg/Kg), the zinc concentrations showed no significant differences between the contaminated TCSFS, BC, and the reference sites OWMA and SNWR. Metal concentrations (Cd, Pb, and Zn) in kidney samples (μg/Kg) are presented by the site in Table 2.

Cadmium concentrations (μg/Kg) in kidney samples from TCSFS, BC and reference sites were significantly different (Table 2). In TCSFS, BC, higher cadmium concentrations (4.62 ± 0.71 μg/Kg) were recorded than at the two reference sites ($P \leq 0.0005$), but the results showed no differences between the two reference sites, OWMA (0.53 ± 0.10) and SNWR (0.53 ± 0.06) μg/Kg. Likewise, Pb concentrations in kidney samples from TCSFS, BC (0.57 ± 0.10 μg/Kg) were higher than in the two reference sites, OWMA and SNWR (0.04 ± 0.01 and 0.05 ± 0.01 μg/Kg respectively).

### Bone microarchitecture relation to heavy metals at TCSFS, BC site

In TCSFS, BC, Cd concentration was positively correlated with trabecular bone separation (r = 0.72, $P \leq 0.05$). Correlation between bone parameters and kidney mineral concentrations by individual site and Pearson's correlation coefficients are presented in Table 4. Cadmium concentrations were negatively correlated with the trabecular bone number (r = −0.67, $P \leq 0.05$), and Pb concentration was positively correlated with trabecular bone separation (r = 0.72, $P \leq 0.05$).

### Bone microarchitecture parameters in relation to heavy metals for uncontaminated sites

Trabecular bone microarchitecture parameters for the lumbar vertebrae (L4) and kidney metal concentrations (Cd, Pb, and Zn) in *Peromyscus leucopus* revealed (1) the differences between the contaminated site TCSFS, BC site (Table 4) and (2) the and the reference sites (*n* = 16) (Table 5). Micro-computed tomography evaluation results of bone parameters showed no correlations with kidney Cd, Pb, and Zn. Kidney lead positively correlated with kidney Zn (0.85, ≤ 0.05).

## DISCUSSION

This environmental toxicology field study showed several impacts and physiological alterations in *Peromyscus leucopus* due to their contact with the contaminants Cd, Pb, and Zn (Tables 1–5). The present study used specimens collected from the TCSFS, BC contaminated area, and two reference sites (OWMA and SNWR). As expected, the concentrations of heavy metals (Cd, Pb, and Zn) in soil and kidney at TCSFS, BC were higher than at the reference sites. This study also analyzed the correlations between mineral concentrations (Cd, Pb, and Zn) of the kidney and the biomarkers such as bone parameters in the biomonitoring species *Peromyscus leucopus*.

Several studies have determined that TCSFS is a site highly contaminated with Cd, Pb, and Zn. Mineral analysis of soil sample results confirmed heavy metal contamination at TCSFS, BC compared to reference sites (OWMA and SNWR). recorded the elevation of cadmium, lead, and zinc in soil sediments at Beaver Creek and Douthat Settling Pond at

TCSFS. Lead concentrations were 440–540 mg/kg and cadmium concentrations were 20–56 mg/kg while zinc concentrations were 3,000–9,300 mg/kg. Large amounts of chat at TCSFS and extensive amounts of Cd, Pb, and Zn from mining and acid water were reported from the 1900s through the 1960s (*Oklahoma Department of Environmental Quality, 2003*). Heavy metals Cd, Pb, and Zn in tailings and yard soil at the Tar Creek National Priorities List Superfund site in Oklahoma were analyzed in order to reduce metal and restore vegetation in this area (*Brown, Compton & Basta, 2007*). Soil chemical analysis showed unequal and extended distribution of heavy metals at the Tar Creek superfund site at chat near Pitcher Oklahoma. The highest concentration of the contaminants was zinc >4,000 ppm, lead >1,000 ppm, and cadmium >40 ppm (*Beattie et al., 2017*).

Bone is one of the most targeted tissues by lead (*Al-Ghafari et al., 2019*; *Lerner, Kindstedt & Lundberg, 2019*; *Yang et al., 2019*). Lead toxicity effects on bone cellular levels cause alterations. These effects include changes in circulating hormone 1, 25-dihydroxy vitamin D3 which regulates bone functions (*Al-Ghafari et al., 2019*; *Lerner, Kindstedt & Lundberg, 2019*). Also, *Martiniaková et al. (2010)* recorded significant heavy metal concentrations in *Apodemus flavicollis* and *Apodemus sylvaticus* at another polluted site in Slovakia. Although slight heavy metal accumulations were recorded in the femora, the study observed no changes in the femora's bone weight and the length of both species.

The present study affirms previous studies which have documented major effects on bone density and osteoporosis resulting from cadmium and lead exposure, including humans (*Youness, Mohammed & Morsy, 2012*). In addition, the findings of this study are also consistent with other studies which have shown that heavy metals can cause liver and renal damage. For example, *Lavery et al. (2009)* investigated heavy metal effects on bone density, other bone parameters, renal damage, and metallothionein (MT) concentrations of South Australian bottlenose dolphins (*Tursiops aduncus*). The results showed Cd, Zn, and Cu in *Tursiops aduncus* caused liver as well as renal damage. Bone parameters of two individuals of *Tursiops aduncus* showed dysfunctions, renal damage, and high levels of MT (*Lavery et al., 2009*).

The findings of this study regarding cadmium contamination at TCSFS, BC have important implications for human health. Bone resorption and negative health effects have been shown to increase in women after middle age due to exposure to even low levels of cadmium in the diet (*Åkesson et al., 2006*). According to a study of women in southeast China in an area heavily polluted by cadmium, cadmium affected bone formation and turnover through indirect effects on vitamin D3 metabolism (*Wang et al., 2003*). Heavy metal toxicity reduces the function of micro and macronutrients such as Zn, phosphate, and calcium which are the main components for bone strength and density.

Generally, this study recorded higher metal concentrations in soil and kidney samples from TCSFS compared with two reference sites. However, the bone microarchitecture analyses of *Peromyscus leucopus* L4 vertebra of contaminated and uncontaminated sites did not show a strong correlation between bone parameters and metal concentration in the kidney. The lack of significant differences could have resulted from a limited sample size ($n = 8$ from each site). Correlations between bone microarchitecture variables appeared to be higher in TCSFS, and BC samples than in reference sites.

In addition, this study used only adult mice, which are more exposed to the contaminants than younger mice due to their age. However, we did not know the exact age of the mice in this study. Bone mineral density and other bone parameter changes can be influenced by age. *Legrand et al. (2000)* recorded several vertebra fractures in a male patient of 52 years of age with lumbar osteopenia. These findings are evaluated through X-ray absorptiometry and bone microarchitecture changes of L2 and L4 trabecular bone. Bone resorption is associated with the inhibition of osteoblast function, and the studies reported this inhibition associated with the lead effects on cellular functions and regulation such as 1, 25–dihydroxy vitamin D3 (*Lerner, Kindstedt & Lundberg, 2019*; *Yang et al., 2019*). Variations in bone parameters were expected due to the specimens' habitat, environmental contaminants, and variable ages.

In the current study, the contaminated site (TCSFS) showed significant elevation in metal concentrations in soil, kidney, and L4 trabecular bone separations of *P. leucopus*. This study showed metal accumulation in a small mammal, *P. leucopus*. The analysis of heavy metal concentrations in the kidney may elucidate issues of environmental quality and physiological alteration.

## CONCLUSIONS

The results of this study reflected actual environmental contamination, but due to the field conditions, this study had several uncontrolled variables such as contaminant levels, duration of exposure to the heavy metals, and animal age. The contaminated site TCSFS, BC location was far from reference sites OWMA & SNWR (SNWR is located substantially further than OWMA from TCSFS). The results are a very important approach to determining the specific endpoint of concern and reaching conclusions that help human health and environmental sustainability. The fact that these mice, *P. leucopus*, had such high mineral concentrations in their kidneys and were still alive raises the possibility that this species, *P. leucopus* adapted to the heavy metal exposure. Moreover, this is the first study to record information regarding bone microarchitecture parameters in *P. leucopus* in North America.

## ACKNOWLEDGEMENTS

I would like to thank the Libyan Ministry of Higher Education and Scientific Research, the Canadian Bureau of International Education (CBIE), and the Libyan-North American Scholarship Program for their assistance in facilitating the study procedures. Great appreciation to Barbara J Stoecker, Ph.D. Departments of Nutritional Sciences and Karen McBee, Ph.D. Integrative Biology at Oklahoma State University. The Environmental Science Graduate Program. Special thanks to Holly Woodward, Ph.D. Oklahoma State University – Center for Health Sciences, and Abdullah Gohar., MSc Mansoura University Vertebrate Paleontology Center (MUVP), Egypt.

### Funding
The author received no funding for this work.

### Competing Interests
The author declares that they have no competing interests.

### Author Contributions
- Maha Abdulftah Elturki conceived and designed the experiments, performed the experiments, analyzed the data, prepared figures and/or tables, authored or reviewed drafts of the article, and approved the final draft.

### Animal Ethics
The following information was supplied relating to ethical approvals (*i.e.*, approving body and any reference numbers):

Oklahoma State University Animal Care and Use provided full approval for this research (Protocol AS056).

### Field Study Permissions
The following information was supplied relating to field study approvals (*i.e.*, approving body and any reference numbers):

Field experiments were approved by the Oklahoma State University Institutional Animal Care and Use Committee (ACUP No. AS066).

### Data Availability
The raw data for the concentrations of zinc, cadmium, and lead in soil samples, body weights and kidney lead, cadmium, and zinc concentrations of *Peromyscus leucopus* samples, and bone microarchitecture are available in the Supplemental File.

### Supplemental Information
Supplemental information for this article can be found online at http://dx.doi.org/10.7717/peerj.14605#supplemental-information.

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
