# Peer review of "Using Peromyscus leucopus as a biomonitor to determine the impact of heavy metal exposure on the kidney and bone mineral density: results from the Tar Creek Superfund Site"

_PeerJ, doi:10.7717/peerj.14605_

## Round 0.1 · original submission · Major Revisions

Dear Authors, Please follow all the requests and suggestions of the reviewers.

Reviewer 1 ·

Basic reporting

No comment

Experimental design

No comment

Validity of the findings

No comment

Additional comments

The article addressed heavy metals contamination, one of the most significant topics that has drawn attention on a global scale. However, the author (s) must carefully consider and resolve a number of notes:
- The introduction should include more contemporary literature on the subject of heavy metals impacts on terrestrial fauna.
- Authors must describe the techniques and sample preservation; also, it is unclear how kidney tissues should be prepared for metal analysis.
- Some sentences in the results section that are more pertinent to methodology are noted in the main text. Additionally, the text does not mention the abbreviations present in the tables, which confuses the reader.
- Discussion , as well as introduction, it needs more comparable findings supported with recent articles.
- Conclusion contains references and a few phrases that would be better placed elsewhere. Reshape the conclusion.
I also made some notes in the main manuscript that the author should take into consideration.

Annotated reviews are not available for download in order to protect the identity of reviewers who chose to remain anonymous.

·

Basic reporting

The study aims to correlate biological variables with soil heavy metal contamination in a field setting.
Despite limitations such as the ignorance about other biological variables (age, nutrition state, heavy metal bones content, etc.), this research represents one of the few studies aiming to use bone architecture as endpoint in a field biomonitoring

The paper need major revision in several parts:

Introduction: Provide more recent literature in the field and specify better the aims of the study
Methods: Be more detailed in the methods and consider reorganizing that section
Discussions: Discuss better the study findings also with more recent literature
Conclusions: More concise, eliminating some parts that are more appropriate in the discussion section

I've provided more specific comments in the PDF

Experimental design

No comment

Validity of the findings

No comment

Additional comments

No comment

---

## Round 0.2 · accepted · Accept

The authors have addressed all of the reviewers' comments.

One previous reviewer accepted the paper and because the other previous reviewer did not respond to the invitation to review the work, I have assessed the revision myself, and I am happy with the current version.

Now the manuscript is ready for publication.

·

Basic reporting

The author made substantial changes to the manuscript
For me is totally acceptable for publication

Experimental design

No comment

Validity of the findings

No comment